# Digital Twin-Based Safety Risk Coupling of Prefabricated Building Hoisting

**DOI:** 10.3390/s21113583

**Published:** 2021-05-21

**Authors:** Zhansheng Liu, Xintong Meng, Zezhong Xing, Antong Jiang

**Affiliations:** Department of Urban Construction, Beijing University of Technology, Beijing 100124, China; mxt0803@163.com (X.M.); xzz1314567@163.com (Z.X.); jiangantong0814@gmail.com (A.J.)

**Keywords:** digital twin, prefabricated building hoisting, hoisting safety, risk coupling, coupling method

## Abstract

Safety management in hoisting is the key issue to determine the development of prefabricated building construction. However, the security management in the hoisting stage lacks a truly effective method of information physical fusion, and the safety risk analysis of hoisting does not consider the interaction of risk factors. In this paper, a hoisting safety risk management framework based on digital twin (DT) is presented. The digital twin hoisting safety risk coupling model is built. The proposed model integrates the Internet of Things (IoT), Building Information Modeling (BIM), and a security risk analysis method combining the Apriori algorithm and complex network. The real-time perception and virtual–real interaction of multi-source information in the hoisting process are realized, the association rules and coupling relationship among hoisting safety risk factors are mined, and the time-varying data information is visualized. Demonstration in the construction of a large-scale prefabricated building shows that with the proposed framework, it is possible to complete the information fusion between the hoisting site and the virtual model and realize the visual management. The correlative relationship among hoisting construction safety risk factors is analyzed, and the key control factors are found. Moreover, the efficiency of information integration and sharing is improved, the gap of coupling analysis of security risk factors is filled, and effective security management and decision-making are achieved with the proposed approach.

## 1. Introduction

The mode of production in the construction industry is characterized by intensive labor input, a large amount of investment, and multi-participation [1]. The construction mode relying on a large number of employees, and the worker’s site construction leads to frequent accidents. The modular construction method of the prefabricated building effectively reduces the pressure of site construction, but there are still serious safety problems. Among them, the most common accidents are in the hoisting stage, such as “falling” and “suffering from mechanical or component collision” [2]. Therefore, the safety risk management in the hoisting of prefabricated buildings cannot be overemphasized. The current research on construction safety management and control mainly focuses on the construction of a safety risk management model and safety risk analysis methods. The safety risk management model refers to the systematic modeling of safety risk management to simulate [3] and optimize [4] the safety problems on the construction site. That is, to simulate the details of the construction and set the safety risk attributes, to analyze the hypothetical construction scenarios, and to most effectively reduce the occurrence of accidents in the construction stage. Ding et al. [5] proposed a web-based safety risk early warning system for subway construction, which adopted a multi-source information fusion model for safety risk management and safety risk early warning. Gunduz et al. [6] set up a dynamic risk management and control model to optimize construction safety management by considering the joint problems among construction stakeholders given the changes in construction conditions. Lin et al. [1] proposed an integrated framework for structural safety closed-loop management based on multi-source data integration to realize visualization of the construction process, generation of the time-varying structural model, and dynamic simulation, monitoring, and evaluation of structural safety. Although, scholars have long found that security risk management has difficulty following the dynamics of the construction site and lacks real-time means of information interaction. It is still difficult to establish an effective information integration and sharing mechanism in the current research. Besides, today’s widely used Building Information Modeling (BIM) can be independently used as a method of security risk management [7]. Some scholars [8,9] use semantic networks and the Internet of Things (IoT) to optimize the integration of the BIM and physical data, but the interaction between BIM model and physical data is still limited to the existing system. It would be better to build a collaborative working framework that avoids silos of information. Digital Twin (DT) can integrate multi-source heterogeneous information to solve the problems of data isolation, fragmentation, and stagnation in the security risk management model [10]. DT and BIM can be combined to improve the efficiency of safety management.

The safety risk analysis method refers to the use of reasonable methods to analyze the safety state of the current construction condition under the assumption that the trend of risk change will remain relatively stable in the short term, so as to infer and predict the occurrence of future accidents [11]. At present, the Fuzzy Analytic Hierarchy Process (F-AHP), Support Vector Machine (SVM), Bayesian Network (BN), and other quantitative methods are mostly used. For example, Gabriel et al. [12] studied the influencing factors of construction risk on site, calculated the index weight and comprehensive risk with an analytic hierarchy process, and evaluated the construction risk on site. Dai et al. [13] proposed a new neural network metamodel for reliability analysis based on wavelet SVM to evaluate the reliability of structures. Zhou et al. [14] put forward the deflection risk analysis model of an underground diaphragm wall based on a Bayesian network to realize the safety risk control in the whole process of subway construction. However, there are some limitations in the methods commonly used at present. For example, F-AHP is complicated to calculate the evaluation of multi-factor and multi-level. SVM is not suitable for large sample data processing. BN needs a lot of data for training and is prone to poor classification accuracy. Moreover, the coupling relationship and interaction among safety risk factors are not considered in the current risk safety analysis and research, so it is difficult to solve the complex coupling problems in the safety management of hoisting construction.

In this study, a DT-based safety risk management framework for prefabricated building hoisting is proposed, and the digital twin coupling model is built emphatically. The coupling relationship among hoisting risk factors is considered, and the hierarchical model of hoisting safety accident occurrence is established in the space-time dimension. Apriori algorithm is used to mine association rules among risk factors of hoisting construction. According to the mining results of association rules, the coupling relationship of safety risk factors in hoisting construction is analyzed by using complex network. The results are correlated with the digital twin model, and the BIM model is used to visually display the information and data of the coupling results, such as reminding the key points of construction safety control, personnel positioning, and site condition inspection. In this way, it can guide the site hoisting construction and improve the data-driven mechanism of the DT-based risk management framework.

The rest of this article is organized as follows. The Section 2 is a literature review. The Section 3 introduces the safety risk management framework of digital twin assembly hoisting and the coupling model part of the framework. The Section 4 describes the method of coupling analysis using a complex network based on mining association rules of Apriori algorithm. The Section 5 is a case study combining a large-scale assembly project to verify the proposed method and framework. Combining the above content, the Section 6 at the end makes a certain summary and outlook.

## 2. Literature Review

This paper presents the design of a DT-based framework for prefabricated building hoisting safety risk management, as well as the construction method of the digital twin security risk coupling model in the framework. This section summarizes the research related to this method in the current literature.

### 2.1. Multisource Data Fusion Framework Based on DT

The concept of Digital Twin (DT) was first proposed by Professor Grieves [15] and applied to the military industry and aerospace fields. As a key enabling technology to solve the problem of physical integration of intelligent manufacturing information and implement the concept and goal of intelligent manufacturing [11], DT has been widely concerned and studied by the academic community, and gradually expand the application field. Tao et al. [16,17] summarized the general digital twin framework and put forward the concept of the DT five-dimensional model by studying the application requirements of complex mechanical and electrical equipment, stereoscopic warehouse, medical treatment, manufacturing workshop, smart city, and other fields. DT also meets the requirements of information and physical data fusion in the field of construction engineering. At present, the research on the application of digital twin theory in the field of architectural engineering has laid a certain foundation. For example, Lu et al. [18] proposed a system framework of digital twin for architecture and urban design, and based on this framework, developed the digital twin model for operation and maintenance management of the West Cambridge site of the University of Cambridge, UK. Liu et al. [19,20] proposed a DT-driven fire safety dynamic evacuation method in the aspect of risk management to realize the three-dimensional model of building safety evacuation, and real-time dynamic planning of an evacuation route according to fire development situation. In the aspect of risk assessment, the safety assessment method of prestressed steel structure in service was studied based on DT technology, and the safety assessment framework of prestressed steel structure based on DT technology was put forward. Dongmin et al. [21] put forward a framework that integrates DT with blockchain to realize information sharing in construction.

To better apply the DT framework in the construction field, it is necessary to find an appropriate information exchange and display platform, so many scholars use BIM for development [22,23,24]. BIM can provide a three-dimensional geometric model of the physical space, which can better simulate the construction site. Meanwhile, BIM can establish the interactive integration mechanism of data information and visualize the perception data and processing data [25]. To improve the efficiency of virtual and real information interaction and better meet the management needs, the realization of BIM visualization in the safety management framework of the hoisting process is usually achieved by extending Industry Foundation Classes (IFC) [26]. The IFC defining the corresponding set of safety attribute and key parameters to achieve the interconnection of the BIM model and the perception data of the framework. After analyzing the perceptual information, management decisions and safety warnings can be made. According to this point, BIM functional model and BIM file database can be established to support the visual display of data information by the BIM model. However, in order to provide more effective and timely support for management decisions, BIM visualization requires real-time perception and interaction of data with new technological means [27]. At present, some scholars [28] combine the Internet of Things (IoT) to carry out more convenient information collection, such as combining Radio Frequency Identification (RFID), wireless communication technology [29,30,31], and intelligent devices [32]. However, the combination of IoT and BIM lacks the function of data mining and analysis. A more ideal way is to use the digital twin framework to realize the fusion of virtual space and physical space. This is also the future model of prefabricated building hoisting safety risk management to be realized.

### 2.2. Security Risk Coupling Analysis

Coupling was originally derived from the concept of physics. In the context of safety risk management, the coupling is interpreted as the degree of interaction between different risk systems or risk factors in complex engineering activities, as well as their inherent correlation [33]. In recent years, the research on the coupling with risk is mainly in the fields of geology, aviation, navigation, etc., and gradually extends to the fields of coal mining and traffic management. For example, Zhang et al. [34,35] used the N-K model to analyze the coupling relationship among risk factors of a gas explosion, and concluded that the probability of gas explosion increases with the increase of the number of risk factors. Xue et al. [36] built a risk coupling model for the risk assessment of high-speed railway projects based on the system dynamics method, which can identify the key coupling effects that increase risks, although their research applied the coupling model for analysis and studied the coupling mechanism of risk accidents. However, the analysis was based on a single risk factor, and the correlation between risks was not analyzed.

The association rule algorithm is used to mine the degree of dependence and association relationship between an event and other events and explore its association rules. The relevant rules can be used to predict future events or the development trend of the system. Therefore, association relationship mining (ARM) is used in risk management and accident analysis. For example, Abhishek et al. [37] used the ARM algorithm to analyze and process the unknown relationship between data, so as to evaluate and manage security risks. Xu et al. [38] used ARM to mine association rules between risk factors and unsafe behaviors of controllers, so as to establish a risk prediction model for unsafe behaviors of air traffic controllers. Apriori algorithm [39] is the most classic one in association rule mining, which takes Support and Confidence as evaluation criteria to mine frequent itemsets of transactions. However, for massive data sets, the mining process may be time-consuming. Therefore, scholars continue to improve the ARM algorithm. For example, Aqra et al. [40] proposed a method to solve the problem of rescanning a previously mined database and allow the retrieval of knowledge that meets multiple thresholds without starting the learning process from scratch. Wen et al. [41] proposed a multidimensional and unconstrained data cube data model to support ARM’s data extraction for analysis requirements. However, the application of ARM in engineering is mainly used in the analysis after the occurrence of accidents. How to apply the association rules mining to the prediction and prevention of risks is a problem that needs to be solved.

The complex network is a method used to show the interwoven relationships within complex systems and to study nodes and relationships between nodes. A large number of complex systems existing in nature and human society can be described by networks, and almost all systems can be abstractly transformed into network models [42]. Therefore, it has aroused wide attention in physics, mathematics, biology, engineering, computers, and other disciplines. To simulate the topology of real networks, scholars [43] have constructed some important network models. Currently, widely used models mainly include random networks, small-world networks, random clustering networks, scale-free networks, and core-edge networks. This also indicates that complex network forms can be used to demonstrate and study the risk-coupled system.

### 2.3. Research Gap

Although DT, risk coupling theory, and association rule algorithm have a certain research foundation in risk management, there is still a lot of research space in the application of hoisting. The research on hoisting safety risks based on DT mainly focuses on risk prediction, but does not really explore the coupling mechanism among the inherent risk factors, and the risk prediction system is not complete enough. Although the application of coupling theory, ARM, and the complex networks has been relatively mature, in the aspect of prefabricated building hoisting, the exploration of coupling theory and the in-depth excavation of association rules among various safety risk factors, security risk are still relatively few. Therefore, this paper proposed a safety risk management framework for prefabricated building hoisting based on DT and built a DT-based coupled model of safety risk factors. The Apriori algorithm is used to mine the association rules among the risk factors. The complex network is used to carry out coupling analysis on the complex system of prefabricated building hoisting risk and identify key control nodes. An effective virtual–reality interaction mechanism for safety risk management in the prefabricated buildings hoisting is established and solves the problem of lack of considering the correlation between factors in hoisting safety risk analysis.

## 3. Framework for DT-Based Safety Risk Management of Prefabricated Building Hoisting

There are many factors affecting the safety risk in the hoisting process of prefabricated building, and all factors are not involved in a single dimension under the development of safety accidents. The traditional safety risk management lacks the analysis of the internal correlation mechanism among the safety risk factors of the assembly building hoisting, and cannot describe its physical scene in a fine way. DT can better realize the fusion of multi-source data and information. According to the general DT framework summarized by Tao et al. [16], the digital twin theory is introduced into the prefabricated buildings hoisting. The safety risk management framework of prefabricated building hoisting driven by DT is formed, and the safety risk coupling model of DT is built.

### 3.1. Framework Overview

As shown in Figure 1, a DT-based safety risk management of prefabricated building hoisting is proposed. The framework uses IoT to connect the hoisting construction site with the hoisting virtual model. Based on high speed, high stability, and low delay data transmission protocols (such as DDS, MQTT, HTTP, etc.), a two-way data synchronous transmission channel between the physical space and the virtual space is built to complete the virtual-real mapping association. The Apriori algorithm is used to mine the association rules among various security risk systems or factors, and the complex network is used to conduct the prefabricated building hoisting risk coupling analysis based on the data mining results. Virtual space modeling is completed based on the BIM model. The BIM model is used as the basis for the visualization of the data service platform in the digital twin framework, which provides functions such as decision making and early warning. In this way, the virtual model and physical site can interact and provide feedback in real-time.

### 3.2. DT-Based Safety Risk Coupling Model

This paper focuses on the construction of the coupling model of DT-based safety risk in the prefabricated building hoisting. By exploring the virtual and real interaction mechanism between the multi-source data of the model and Digging deep into association rules among security risk factors., the coupling effect of safety risk factors is analyzed. In this paper, the information described by the coupling model of assembly hoisting safety risk is defined from five dimensions. The coupling model can be defined as $M_{DT}$, which is shown in Formula (1):(1)$$M_{DT}=\left(PH,VH,RS,CN,DD\right)$$
where: PH refers to the real prefabricated building construction hoisting site; VH refers to the virtual information model of prefabricated building hoisting, which relies on the BIM model for visual display, and combined with other risk information related models; RS refers to hoisting safety risk service, which include RS_c and RS_s; coupling service RS_c provides an inherent analysis of security risks; the analysis of risk factors based on RS_c supports the implementation of service RS_s; CN refers to the connection between various parts of the $M_{DT}$; DD refers to the twin data, obtained by fusing physical data with simulation data and feedback data. The model is shown in Figure 2 in conjunction with an illustration.

In this model, the coupling service RS_c first uses the Apriori algorithm and complex network to analyze the coupling relationship of the collected historical data. The analysis results will be used as the initial rule model to drive service RS_s and stored in the virtual hoisting model VH. RS_s guide the physical hoisting site PH according to the analysis results. When the hoisting work begins, the twin data DD formed by the fusion of physical data and virtual data will be generated. RS_s will continue to optimize and update the initial association rule model based on DD, and promote VH to better characterize PH. The data interaction and feedback among the dimensional modules in the model are completed by connection CN.

### 3.3. Perception and Interaction of Data

In the coupling model of prefabricated building hoisting safety risk based on DT, it is necessary to grasp the spatiotemporal development of each risk factor in the hoisting process in real-time. Therefore, multi-source risk factor information data should be perceived and integrated into the process of prefabricated building hoisting. Due to the data requirements of this method, a self-organizing Wi-Fi network (MESH) distributed IoT structure is adopted in this paper to realize data perception. For this form of IoT structure, Ding et al. [44,45] have conducted some research and exploration in lifting safety warning and blind lifting safety monitoring of underground buildings. The network structure is composed of two parts: a sensor node and a remote intelligent control center which can communicate wirelessly. Several sensor nodes can flexibly form a small subnet according to the assignment of tasks, and each subnet is integrated into the control center. The communication between each sensor or subnet and the control center depends on the Network Interface Card (NIC) based on wireless broadband. The analysis and storage of data information rely on the cloud server. For example, ref. [45] selected a global cloud computing company (Hangzhou Alibaba Cloud Server) to store the database.

In the coupling model of prefabricated building hoisting safety risk based on DT, the risk factor indicators of assembly hoisting that need to be collected will be analyzed in Section 4.2. For the information to be collected, four types of acquisition terminals will be used in this paper: (1) Smart camera—recording the wearing of safety equipment, and providing real-time video data on the construction site. (2) Environmental sensors—including wind speed, temperature, humidity, air quality, and other sensors. It is used to record environmental conditions. (3) Tower data recorder—monitoring and recording the working condition of the tower crane. (4) RFID tag—it is used to perceive and grasp the state information of prefabricated components. RFID tags are pasted or installed on the surface of the component, and the information is collected by its special scanner. In addition, the collection of management record information depends on the real-time upload of information by the management personnel using mobile devices to complete the collection.

To ensure the real-time transmission of data, to prevent the signal being blocked, resulting in data transmission delay. As shown in Figure 3, the self-organizing Wi-Fi network equipment can be used for pairwise free networking. When a certain line is interrupted, data can be transmitted through other links. The data will be uploaded to the cloud server for analysis and storage, and then transferred to the command center for application and assignment of new tasks. For example, when the connection between the environmental information collection terminal and the Mesh network center is damaged, the subnet can be formed by combining with the management information uploading terminal, and the subnet information can be transmitted to the network center through the line of the management information uploading terminal.

The data types in the coupling method can be divided into structured data, unstructured data, and coupled (after processing) data. The data collection process and the parts contained in the database are shown in Figure 4. Data collected by the environmental sensor and tower crane recorder are structural data, which are stored in the structural database. Data information such as video, image, and data records formed by intelligent monitoring and management records, as well as information such as component state, belong to the unstructured data and are stored in unstructured database. After data coupling analysis, the coupling result data is stored in the coupling database.

## 4. Safety Risk Coupling Analysis Method of Prefabricated Building Hoisting

### 4.1. Safety Risk Coupling Mechanism

Hoisting is an important part of prefabricated building construction, but there is still a lot of exploration space for safety risk analysis of the hoisting process. The occurrence of assembly hoisting safety accidents can be based on the Critical State Theory. Based on the accident cause model, the spatial dimension is added based on [46], and the three-dimensional risk accident system that evolves in space at any time is considered. As shown in Figure 5, the occurrence of hoisting safety accidents can be divided into different levels in the mode of point, chain, network, and layer. A multi-dimensional system is established from the link layer, and factors at all levels develop in accordance with time, space, and coupling levels [46]. The bottom layer of human, machine, material, pipe, ring five single safety risk factor system. Each of these systems includes a number of individual risk factors. When a certain risk state is triggered, the underlying risk factors are coupled to strengthen or weaken each other. According to the path of safety risk factor → safety risk accident subsystem → safety risk accident system, the transition occurs to the upper level. Finally, breaking the critical point evolves to form the hoisting safety accident.

Analyzed from the perspective of correlation, the coupling and superposition of risk factors will show zero coupling, weak coupling, and strong coupling [12]. If the risk index layer is $R=\left\{r_{1},r_{2},r_{3}\cdots r_{n}\right\}$ and the risk accident system layer is $S=\left\{s_{1},s_{2},s_{3}\cdots s_{i}\right\}$, then its three coupling states can be well expressed as:

(1)
$S\left(t\right)=\max\left\{r_{1}\left(t\right),r_{2}\left(t\right),r_{3}\left(t\right)\cdots r_{n}\left(t\right)\right\}$


That is the state of “zero coupling”, the superposition of multiple influencing factors does not change the security risk state caused by the most influential factor:

(2)
$S\left(t\right)<\max\left\{r_{1}\left(t\right),r_{2}\left(t\right),r_{3}\left(t\right)\cdots r_{n}\left(t\right)\right\}$


That is the “weak coupling” state. After multiple influencing factors are coupled, the effect of the most influential influencing factor in the original combination is weakened, and the resulting security risk state is less than the influencing effect of its largest influencing factor.

(3)
$S\left(t\right)>\max\left\{r_{1}\left(t\right),r_{2}\left(t\right),r_{3}\left(t\right)\cdots r_{n}\left(t\right)\right\}$


That is the “strong coupling” state. Under the coupling effect, the influence of multiple influence factors on the final security risk is increased, which is greater than the influence effect of the largest influence factor, but usually does not exceed the complete superposition of the influence factors.

The coupling part of this method mainly solves the “strong coupling” state in the safety risk of prefabricated buildings hoisting. The implementation process of the coupling analysis method is shown in Figure 6. Firstly, the safety risk factors in the hoisting stage of the prefabricated building are identified and set up a security risk indicator system. Then, the Apriori algorithm is used to mine association rules among hoisting safety risk factors. Based on the ARM results, the characteristic variables of the complex network are extracted. To construct a complex safety risk network for prefabricated building hoisting. The coupling effect among security risk factors is analyzed by using the complex network to find out the key influencing factors. The follow-up twinning service can timely control the key factors, reduce the chain of security risk accidents, and prevent the formation and evolution of security risk networks.

### 4.2. Security Risk Coupling Analysis

#### 4.2.1. Data Processing

The ARM carried out by the Apriori algorithm is aimed at single-dimensional and single-layer data, while the safety risk data of prefabricated building hoisting present multi-dimensional and multi-layer characteristics, so it is necessary to process the data to some extent. According to the hierarchical division in Section 4.1, the study was conducted through engineering practical experience and literature [47,48,49]. The main safety risk factors of assembly hoisting are classified and summarized. The results are shown in Table 1. The data were modeled according to the classification results of risk factors so as to import the risk coupling model for coupling analysis. The form of data set is shown in Formulas (2)–(4):(2)$$S=\left\{s_{1},s_{2},s_{3}\cdots s_{i}\right\}$$
(3)$$S=\left\{s_{ub1},s_{ub2},s_{ub3}\cdots s_{iubm}\right\}$$
(4)$$R=\left\{r_{1},r_{2},r_{3}\cdots r_{n}\right\}$$
where: the prefabricated building hoisting safety risk accident system layer is recorded as S; which is a collection of specific risk index $s_{i}$; the hoisting safety risk accident subsystem layer is recorded as $S_{ub}$, and its risk index is marked as $s_{ub}$; the safety risk factor layer is recorded as R, and the risk index is defined as $r_{n}$; and the safety risk accident layer is recorded as A.

Risk factor indicators are divided into state variables and continuous variables. To extract frequent itemsets more conveniently, data need to be simplified and integrated. The risk status classification standard of the two variables and the risk accident grade classification standard are established.

(1)Risk factor status classification

Referring to expert experience and literature as well as engineering investigation and practice, the criteria for dividing the risk status of discrete variables into two-states and three-states are shown in Table 2 and Table 3. The division of continuous variable risk status is shown in Table 4.

In Table 4, $ST_{1}$ indicates good condition. $ST_{2}$ means normal condition. $ST_{3}$ is in poor condition. In addition, when mining association rules, to output risk indicators and status intuitively, risk states are first corresponding to risk indicators. For example, “$ST_{1}$—$r_{4}$” means “the rate of personnel participating in safety clarification is in good condition”.

(2)Risk classification

Due to the different nature and value range of data, it is difficult to classify the risk level, so the data is firstly normalized. That is, the value after processing is reserved between [0, 1] [50]. Firstly, the discrete variables of two-state and three-state were transformed into continuous variables by means of expert rating. A three-state discrete variable is transformed into “$ST_{1}$—[90, 100), $ST_{2}$—[60, 90), $ST_{3}$—[0, 60)”. Two-state discrete variables are transformed into “$ST_{1}$—90, $ST_{2}$—50”. After the transformation is completed, normalization is carried out according to the coordination relationship among the indicators, and its calculation formula is shown in formula (5):(5)$$u_{i}=\frac{x_{i}-m_{i}}{n_{i}-m_{i}}$$
where: $x_{i}$ is the observed value of variable $r_{i}$; $u_{i}$ is the observed value of this variable after normalization; the value of this variable ranges from $ST_{1}$ to $ST_{3}$ with a minimum value of min=m and a maximum value of max=n.

After normalization, the risk level was evaluated by the size of the $\sum_{i=1}^{n}u_{i}$ value. The higher the value, the higher the risk level. The classification of risk accident levels is shown in Table 5:

#### 4.2.2. Association Rules Mining

The Apriori algorithm is used to refine the common characteristics of the factors that lead to the occurrence of a certain type of accident and explore the concomitant effects and association rules among the security risk factors. The Apriori algorithm uses the framework of “support-confidence” to calculate and analyze the probability. The frequent itemsets is obtained by specifying the minimum support (Minsup), and the minimum confidence threshold (Minconf) is specified to mine the hidden relationship between risk data.

Firstly, the data are sorted out and recorded. The safety risk factor index $r_{n}$ was combined with its safety state $ST_{1}$. According to the calculation rules of the Apriori algorithm, when an index is in a certain state, it is counted as 1 in the corresponding state and 0 in the other state. Frequency is the total number of times of this indicator in this state. Form a list of records, and then normalize each data record. According to the value of $\sum_{i=1}^{n}u_{i}$, each record corresponds to three risk accident levels: $s_{1}$, $s_{2}$, $s_{3}$, and divides the list into three parts. Mining association rules between $r_{na}$ and $r_{nb}$ in each part, respectively. The mining process is carried out according to formulas (6) and (7):(6)$$Support\left(r_{na}\Rightarrow r_{nb}\right)=Support\left(r_{na}\cup r_{nb}\right)=P\left(r_{na}r_{nb}\right)\;$$
(7)$$Confidence\left(r_{na}\Rightarrow r_{nb}\right)=Support\left(r_{na}\cup r_{nb}\right)/Support\left(r_{na}\right)=P(r_{na}|r_{nb})\;$$
where: $Support\left(r_{na}\Rightarrow r_{nb}\right)$ is the rule support degree, which means the probability that the hoisting safety risk event $r_{na}$ and the risk event $r_{nb}$ occur at the same time; $Support\left(r_{na}\Rightarrow r_{nb}\right)$ is the confidence degree, which means the probability of the event $r_{nb}$ occurring when the risk event $r_{na}$ occurs; when $Confidence\left(r_{na}\Rightarrow r_{nb}\right)$ is greater than Minconf, the association rule between these two events is considered strong; that is, security risk events $r_{na}$ and $r_{nb}$ always occur at the same time, or when risk event $r_{na}$ occurs, it will always be accompanied by risk event $r_{nb}$.

Although Apriori algorithm has high advantages in association rule mining, the traditional Apriori algorithm still has many disadvantages, such as too many intermediate item sets, frequent database scanning and unique support. These shortcomings may lead to some misleading mining results. A new parameter Lift is usually introduced to improve the traditional Apriori algorithm. The calculation method is shown in formula (8):(8)$$Lift=Confidence\left(r_{na}\Rightarrow r_{nb}\right)/Support\left(r_{na}\Rightarrow r_{nb}\right)$$
where: when Lift<1, the data association relationship is regarded as meaningless and will not be analyzed in the results; at the same time, Lift can be used to represent the association strength of the left and right sides of the association rule.

The number and quality of association rules mined by the Apriori algorithm are determined by the selected support and confidence thresholds. Take association rules mining for extremely high-risk state ($s_{1}$) as an example. Under the selection of different thresholds of minimum confidence and minimum support, the number of association rules is shown in Figure 7. Extreme cases such as “0.1–0.1” and “1–1” will cause significant errors when selecting parameters, and it should be selected from a gentle stage [50]. In this case, the parameter can be selected as Support=0.5,Confidence=0.6.

#### 4.2.3. Complex Network Analysis

The complex network can well show the relationship between nodes and the degree of influence and dependence on each other. Therefore, it has an outstanding advantage to use a complex network to analyze the coupling relationship between risk factors and find the key nodes in the coupling network. At the same time, a complex network provides a way to visualize the coupling relationship.

The complex network has the same definition as graph theory. Its topological structure determines the function of the whole system, but different from graph theory, the complex networks can analyze the dynamic changes of network systems. According to the topology of the complex network, the characteristic variables of the network are extracted based on the results of ARM, and the complex networks are built. The coupling effect of hoisting safety risk in prefabricated building is analyzed and visualized. As shown in Figure 8, the risk factor index is the node of the complex network, the edge represents the existing correlation between various factors, and its weight represents the degree of dependence among factors. Therefore, its nodes are indicators of security risk factors corresponding to the risk state on the left side of association rules. Its edges are association rules between indicators. The weight is the correlation degree, that is, the value of Lift after ARM deduction.

The complex network can also be understood as a group of nodes connected by edges. Its main statistical characteristics include degree ($D_{i}$), average shortest path length (ASPL), and betweenness centrality ($BC_{i}$). In this paper, the main degree of statistics and analysis, in a Graph, the degree of node i indicates the $D_{i}$ of a direct association between the node and other nodes in the network. The greater the degree of a node, the more central the node is in the network. The calculation method of $D_{i}$ is shown in formula (9):(9)$$D_{i}=\overset{g}{\underset{j=1}{\sum }}x_{ij}$$
where: $D_{i}$ represents the centrality of node i; i and j represent different nodes that are connected to each other in the network; i≠j. $x_{ij}$ represents the edge between nodes i, j; according to the statistical characteristics and network structure, the coupling relationship of assembly hoisting safety risks can be analyzed, and the key nodes can be obtained.

## 5. Case Study

### 5.1. Project Background

Building 10#~18#, part of supporting buildings, and part of an underground garage of a large prefabricated engineering project are all prefabricated building structures. A lot of hoisting work of prefabricated components is involved in the project construction. It includes the hoisting of precast stairs, precast wall panels, and composite floor slabs. Moreover, some BIM models have been established for the project. The project plan schematic diagram, prefabricated components, and the corresponding BIM model are shown in Figure 9. Taking it as a case to study the related content of prefabricated building hoisting has a significant representative value.

### 5.2. Framework Implementation

#### 5.2.1. Data Preparation

This paper focuses on the exploration of a safety risk coupling method for prefabricated building hoisting based on DT. Therefore, the significance of the case study in this paper lies in further clarifying the procedure of the method and verifying the feasibility of the method. Due to the engineering data itself, it has a rather individual case. The results of data mining analysis cannot apply to other engineering projects. Data to be prepared by this method includes two parts: the historical empirical data to form the initial rule model and the actual data of the project. The data sources of the two parts in this case are as follows:

(1) Collecting the hoisting accident situation according to the data collection steps in the literature [47], and consulting experts after sorting out the results, forming one historical record corresponding to the risk state classification criteria in Section 4.2.1.

(2) The actual engineering data part comes from the data perception and safety management records of the construction site. The data perception method in Section 3.3 is applied to collect the site construction data, and the collected data is recorded according to the risk status classification criteria in Section 4.2.1. As this data acquisition method is performed in real time, the repeatability of data recorded at each moment in the same day is high. To verify this method more effectively, the data records of each day were integrated to form one valid record. The records of 26 consecutive days were selected from all records since the application of the framework of the project as part of the actual engineering data.

The data of the two parts were integrated and used as the data source to verify the method. The data recording method and partial results are shown in Table 6. Firstly, the sorted data are normalized according to the calculation method in Section 4.2.1, and calculated $\sum_{i=1}^{n}u_{i}\;$ for each record. The record corresponds to safety accident levels $s_{1}$, $s_{2}$, $s_{3}$ according to the value of $\sum_{i=1}^{n}u_{i}$. The data were divided into general risk data group, high risk data group and extremely high risk data group to prepare for the mining of hoisting safety risk association rules in the following steps.

#### 5.2.2. Construction of Coupling Model

(1)Data mining

The Apriori algorithm part of this method is implemented in Python. The divided data set is read, and the association rules among the three risk factors are mined. The parameters of the Apriori algorithm are selected according to the previous Section 4.2.2. That is, the rule at Support=0.5,Confidence=0.6 is used as the output value of the association rule. The value of Lift is used as a parameter to measure the strength of the rule. Among the 1467 association rules generated, the mining results of the six association rules with the highest promotion degree are shown in Table 7.

In the mining results of association rules in the table, the relationship between the left side and the right side of the association rules is expressed as: the combination of security risk indicators with risk status on the left side of the association rules is very likely to cause a certain level of risk accidents on the right side. In the aspect of hoisting safety management, the element on the left is the object that needs to be paid attention to in the safety risk management, and the element on the right is the risk accident level easily caused by this combination of risk factors. Lift is the correlation strength between the combination of safety risk factors and the resulting accident level. For example, the first rule in the table: “{Large acceleration, Illegal operation, Poorer ratio of personnel participating in safety confession} → {Higher risk $s_{2}$}, Lift=1.7994”, indicating that when the tower crane is hoisted or dropped, the acceleration is large, and the personnel operates illegally. Additionally, when the proportion of people participating in safety confession is low, it has a higher risk and is prone to larger safety accidents when it occurs at the same time. By consulting relevant personnel to review the project history and inspect the site, the rules are effective.

(2)Complex network establishment

According to the establishment method of the complex network of security risks described above, the eigenvalues of the complex network were extracted based on the mining results of the association rules of the hoisting security risks of the prefabricated construction project, and the ARM results were mapped to the complex network. The generated complex network of high risk conditions and extremely high risk conditions is shown in Figure 10. The weighted network takes full account of the interaction among the combinations of safety risk factors and demonstrates the coupled accident chain.

The statistical characteristics of the complex network are calculated to find the key risk nodes. In the statistical characteristics of complex networks, the higher degree of nodes indicates that they have higher influence ability in the coupling system, that is, they are in a more critical position. The characteristic values of the complex network sorted according to its degree ($D_{i}$) are shown in Figure 11. Among them, the node {violation—whether the operation violates the regulations}, {poor—safety protection wearing state}, {low—participants rate of safety briefing}, {longer—actual service life of hoisting machinery}, {high—actual load ratio} has a relatively high eccentricity, and it is at a key position in the network and should be controlled during the hoisting process. In addition, {larger—wind speed at the hoisting construction site} and {larger—acceleration} are also prominent. As a secondary key node, attention should be paid to the construction.

#### 5.2.3. On-Site Construction Guidance

After completed the construction of the prefabricated hoisting security risk coupling model and maps, the coupling results back to the digital twin model. Through the above-mentioned wireless ad hoc network Internet of Things connection mode, the field data is sensed and updated, and the result information is transmitted to the lifting control platform after the coupling model is analyzed. The platform is developed based on the BIM model. First, the BIM model of the hoisting site is built, and the BIM model is used to fully simulate the site by defining the parameters and attribute sets. The BIM data is interconnected with the rest of the framework through the IFC standard to realize the mutual transmission of data information. Through preset decision-making and early warning and other functional modules, combined with the BIM model for visual display on the computer and mobile terminals, it is convenient for managers to guide on-site construction based on the platform. For example, when a new record is generated and the security level of the record is calculated to be $s_{3}$, it will be indicated in the platform that a great risk will occur and the key factors with high degree will be indicated in the coupling results. Managers should check and control these key factors in a timely manner to destroy the coupling network and prevent the evolution of coupling risks. In addition, each coupling result and operation probability will be used as the basis for the next update, and the coupling data of each update will be backed up to the data cloud platform for storage for review and system maintenance.

## 6. Conclusions and Future Works

The current research on construction safety management and control mainly focuses on the construction of a safety risk management model and safety risk analysis methods. In the aspect of the establishment of a security risk management model, the difficult problem of security risk management lies in the difficulty of establishing effective information integration and sharing mechanism. Due to the lack of real-time information interaction means, it cannot follow the dynamics of the construction site well. In the research of safety analysis methods, the common methods do not consider the coupling relationship and interaction among safety risk factors. It is difficult to solve the complex coupling problems in hoisting construction safety management.

In this paper, a safety risk management framework for prefabricated building hoisting based on DT is proposed. On the basis of this framework, the construction method of the DT coupling model is studied. The model integrates the Internet of Things (IoT) and Building Information Modeling (BIM), as well as the security risk analysis method combining the Apriori algorithm and complex network. Considering the coupling relationship between hoisting risk factors, the hierarchical model of hoisting safety accidents in space-time dimension is established. Apriori algorithm is used to mine association rules among risk factors of hoisting construction. According to the mining results of association rules, the coupling relationship of safety risk factors in hoisting construction is analyzed by using a complex network. The BIM model is used to visually display the data information, so as to guide the site hoisting construction. The methods have been validated and tested in a large-scale prefabricated construction projects, proving the value of the framework, and the effectiveness of the diagnostics methods. The main contributions of this study are as follows:

(1) DT can improve the safety management method and mode of construction. This paper further promotes the development of DT in the field of prefabricated building hoisting safety management and improves the data-driven mechanism of hoisting safety risk management framework based on DT. DT also has a high application value in other aspects of the engineering construction field. This framework can also be extended to other aspects of prefabricated building management.

(2) A safety risk coupling method for a safety risk coupling method for prefabricated building hoisting based on DT is proposed. The interaction mechanism between physical information and virtual information in the framework is described. The efficiency of information integration and sharing is improved. It fills the vacancy of coupling analysis of security risk factors. Effective safety management and decision-making are realized.

This study provides a good method to integrate multi-source heterogeneous information in the process of hoisting safety risk management of prefabricated buildings. Solve the problems of data isolation, fragmentation, and stagnation in the security risk management model. There are still some defects in this paper, such as an insufficient collection of validation data samples and lack of an intelligent evaluation system for the rules mined. In addition, how to integrate this method with other prefabricated buildings hoisting safety twin services also needs to be further explored.

## Figures and Tables

**Figure 1 sensors-21-03583-f001:**
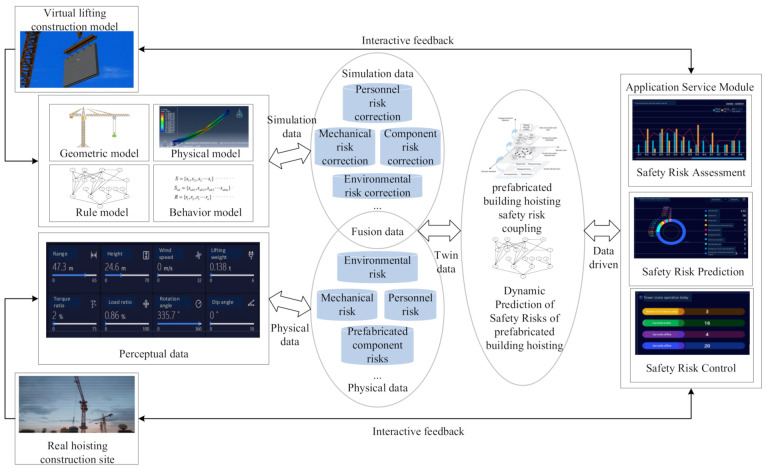
Framework for DT-based safety risk management of prefabricated building hoisting.

**Figure 2 sensors-21-03583-f002:**
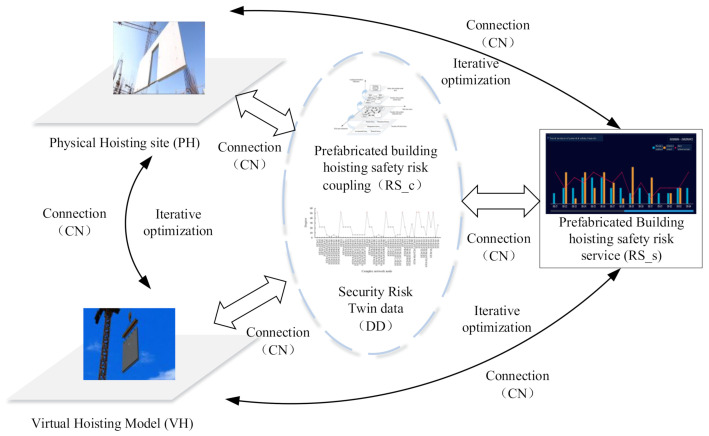
DT-based safety risk coupling model of prefabricated building hoisting.

**Figure 3 sensors-21-03583-f003:**
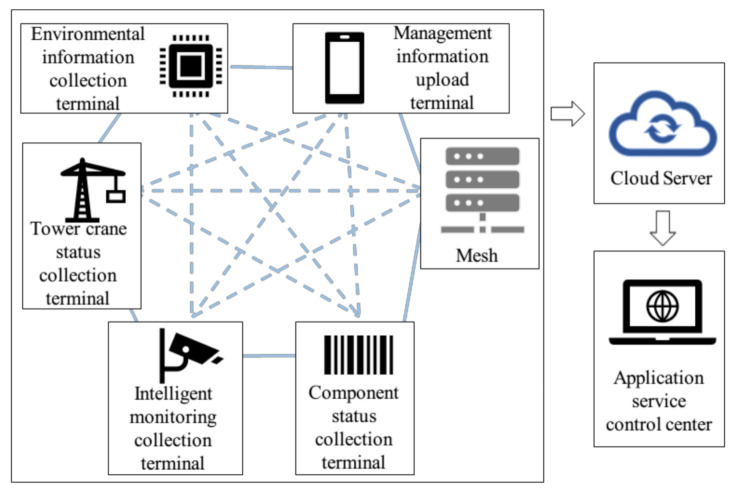
The architecture of IoT system with a self-organizing Wi-Fi network.

**Figure 4 sensors-21-03583-f004:**
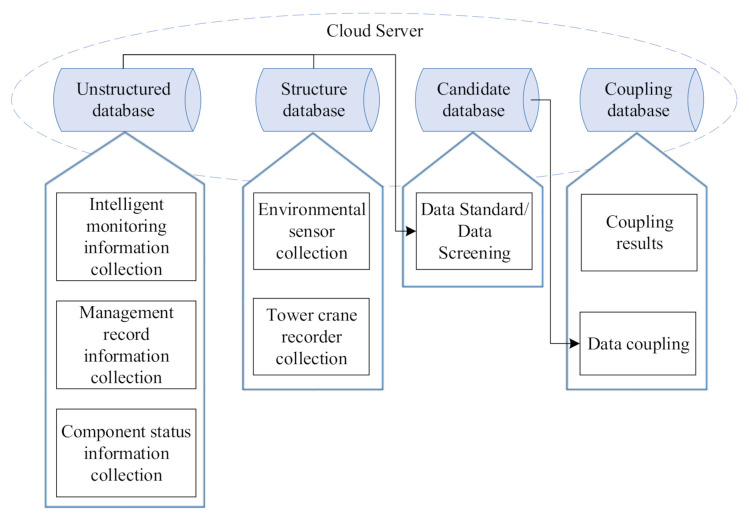
Data storage structure in the IoT system with a self-organizing Wi-Fi network.

**Figure 5 sensors-21-03583-f005:**
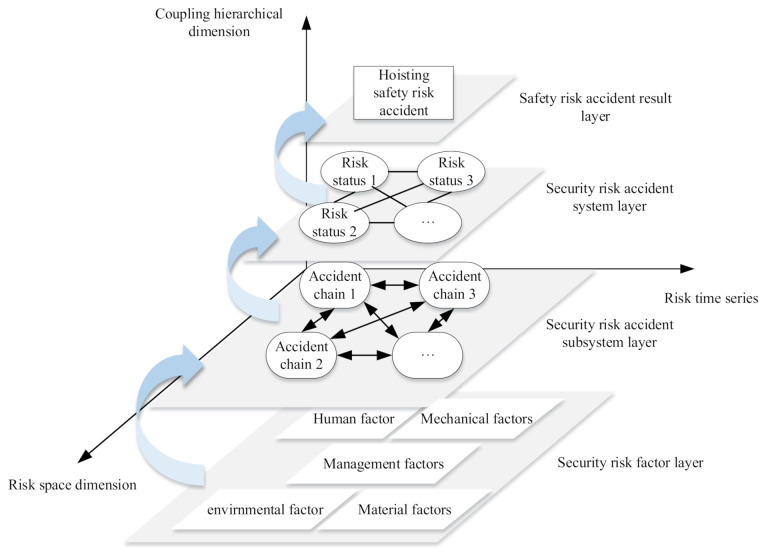
Coupling level diagram of safety risk accidents in prefabricated building construction.

**Figure 6 sensors-21-03583-f006:**
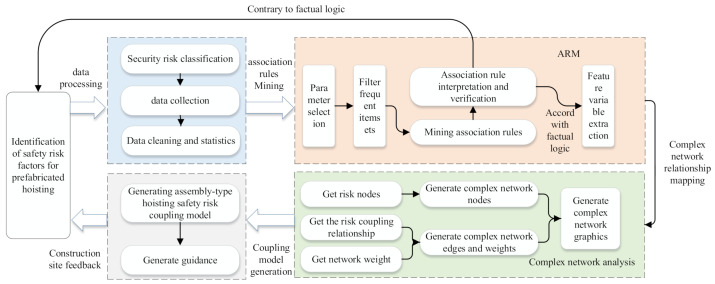
The process of establishing the safety risk coupling model of the prefabricated building hoisting.

**Figure 7 sensors-21-03583-f007:**
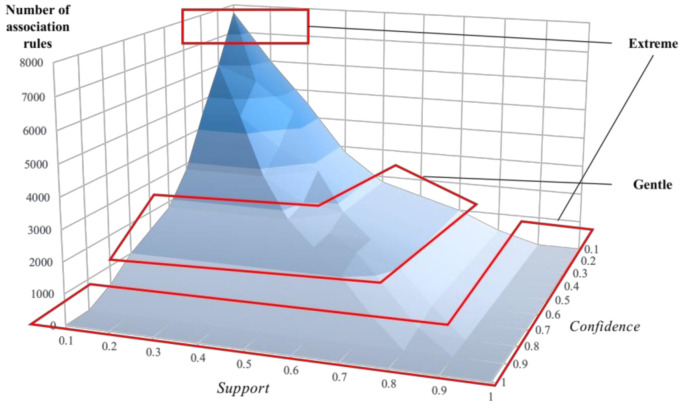
Influence of different parameters values on the number of association rules.

**Figure 8 sensors-21-03583-f008:**
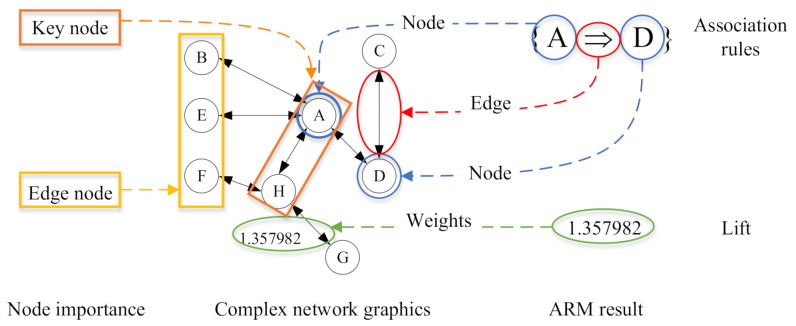
Mapping ARM results to complex networks.

**Figure 9 sensors-21-03583-f009:**
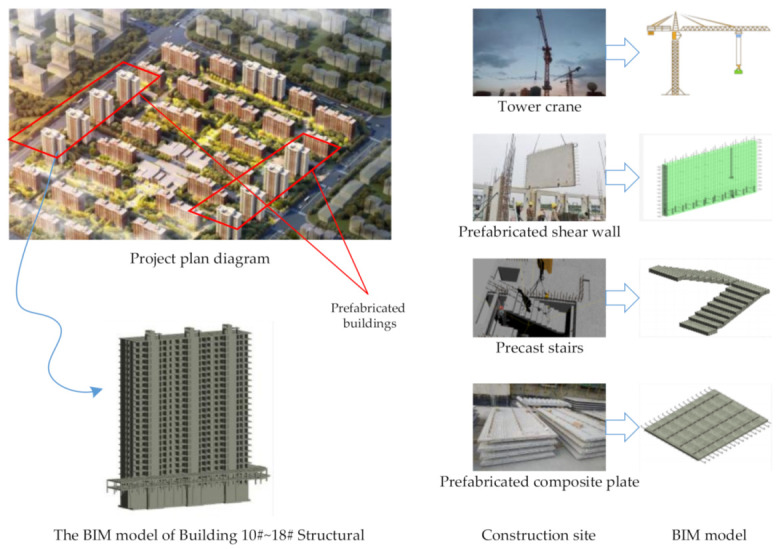
The project plan schematic diagram, prefabricated components, and the corresponding BIM model.

**Figure 10 sensors-21-03583-f010:**
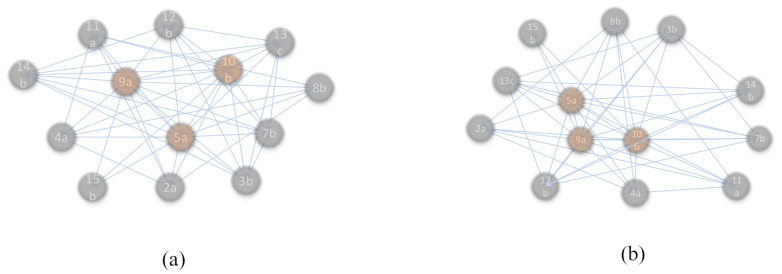
Safety risk complex network of prefabricated building hoisting. (**a**) A Complex network of coupled system in a high risk state; (**b**) a complex network of coupled system in an extremely high risk state.

**Figure 11 sensors-21-03583-f011:**
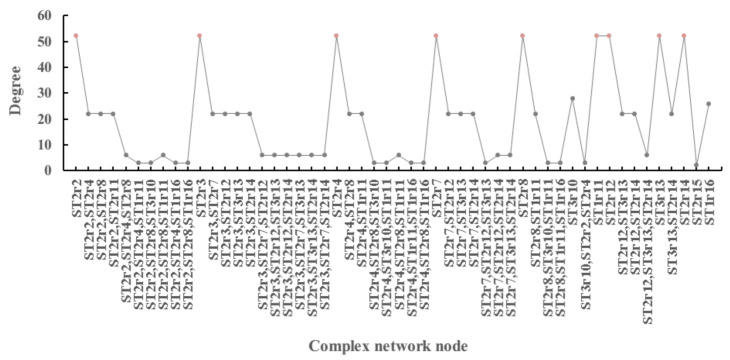
Complex network degree ($D_{i}$) statistics of coupling system.

**Table 1 sensors-21-03583-t001:** Safety risk classification of the prefabricated building construction hoisting process.

Number	Level	Coding	Specific Risk Indicators	Coding
1	Security incident layer	A	Lifting safety accident	A
2	Hoisting safety risk accident system layer	S	General risk $s_{1}$; high risk $s_{2}$; extremely high risk $s_{3}$	$s_{i}$
3	Lifting safety risk accident subsystem layer	$S_{ub}$	Human factor accident chain; management factor accident chain; environmental factor accident chain; mechanical factor accident chain; material factor accident chain	$s_{ub}$
4	Security risk factor layer	R	Technical level of operators $r_{1}$; operational violations by operators $r_{2}$; safety protection wearing status $r_{3}$; percentage of people participating in safety clarification $r_{4}$; sling inclination $r_{5}$; quality level of prefabricated components $r_{6}$; actual service life of hoisting machinery $r_{7}$; wear rate of hoisting equipment $r_{8}$; speed $r_{9}$; acceleration $r_{10}$; prefabrication rate $r_{11}$; cross-interference situation of tower crane operation $r_{12}$; actual load ratio $r_{13}$; construction safety management level of hoisting site $r_{14}$; security measures cost investment ratio $r_{15}$; wind speed of hoisting construction site $r_{16}$; layout of prefabricated components storage yard $r_{17}$	$r_{n}$

**Table 2 sensors-21-03583-t002:** Two-state discrete variable risk state division.

Risk Indicators	$ST_{1}$	$ST_{2}$
Operational violations by operators ($r_{2}$)	Compliance	Illegal operation
Safety protection wearing status ($r_{3}$)	Complete	Missing accessories
Cross-interference situation of tower crane operation ($r_{12}$)	Uncrossed	Cross

**Table 3 sensors-21-03583-t003:** Three-state discrete variable risk state division.

Risk Indicators	$ST_{1}$	$ST_{2}$	$ST_{3}$
Technical level of operators ($r_{1}$)	Good	General	Poor
Quality level of prefabricated components ( $r_{6}$ )	Good	Qualified	Unqualified
Construction safety management level of hoisting site ($r_{14}$)	Good	General	Poor
Layout of prefabricated components storage yard ($r_{17}$)	Good	General	Poor

**Table 4 sensors-21-03583-t004:** Continuous variable risk state division.

Risk Indicators	$ST_{1}$	$ST_{2}$	$ST_{3}$
Percentage of people participating in safety clarification ($r_{4}$/%)	[90, 100)	[60, 90)	[0, 60)
Sling inclination ($r_{5}$/°)	[30, 40)	[40, 50)	[50, 60)
Actual service life of hoisting machinery ($r_{7}$/a)	[0, 5)	[5, 10)	[10, 20)
Wear rate of hoisting equipment ( $r_{8}$ /%)	[0, 10)	[10, 40)	[40, 50)
Speed ($r_{9}$/m·s^−1^)	[0, 40)	[40, 60)	[60, 80)
Acceleration ($r_{10}$/m·s^−2^)	[0, 0.015)	[0.015, 0.025)	[0.025, 0.045)
Prefabrication rate ($r_{11}$/%)	[0, 30)	[30, 50)	[50, 100)
Actual load ratio ( $r_{13}$ /%)	[0, 80)	[80, 100)	[100, 150)
Security measures cost investment ratio ( $r_{15}$ /%)	[3, 5)	[1.5, 3)	[0, 1.5)
Wind speed of hoisting construction site ($r_{16}$/m·s^−1^)	[0, 7.9)	[7.9, 10.8)	[10.8, 16)

**Table 5 sensors-21-03583-t005:** Classification standard of safety risk levels.

Risk Accident Level	$s_{1}$	$s_{2}$	$s_{3}$
Normalized value	[0, 0.243)	[0.243, 13.275)	[13.275, 17)

**Table 6 sensors-21-03583-t006:** Continuous variable risk state division.

	Risk Indicators	$r_{1}$			$r_{2}$		$r_{3}$		$r_{4}$			S		
Record the Moment		$ST_{1}$	$ST_{2}$		$ST_{1}$	$ST_{2}.\;$	$ST_{1}$	$ST_{2}$	$ST_{1}$	$ST_{2}$	$ST_{3}$	$s_{2}$	$s_{3}$
Moment 1	1			1		1		1			1		
Moment 2			1		1		1			1			1
Moment 3		1		1			1	1				1	

**Table 7 sensors-21-03583-t007:** Mining results of association rules with higher promotion degree.

Number	Left		Right	Lift
1	{Higher acceleration, illegal operation, poorer ratio of personnel participating in safety confession}	→	{$s_{2}$}	1.7994
2	{Higher acceleration, average prefabrication rate, poor hoisting equipment wears}	→	{$s_{2}$}	1.7893
3	{Higher acceleration, illegal operation, poor hoisting equipment wears}	→	{$s_{2}$}	1.756
4	{High wind speed, illegal operation, poorer ratio of personnel participating in safety confession}	→	{$s_{3}$}	1.7508
5	{High wind speed, poorer ratio of personnel participating in safety confession, poor hoisting equipment wears}	→	{$s_{3}$}	1.7428
6	{High wind speed, higher acceleration}	→	{$s_{3}$}	1.7393

## Data Availability

Data sharing not applicable.

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
