# Peer review of "Digital Twin-Based Safety Risk Coupling of Prefabricated Building Hoisting"

_sensors, 2021, doi:10.3390/s21113583_

Round 1
Reviewer 1 Report
The work proposed a safety risk coupling method for assembly and hoisting based on digital twins, providing guidance for the execution of other twin services such as security risk assessment, prediction and control for engineering projects. This method considers the coupling relationship analysis among various risk factors, and puts forward association rules for risk safety prediction, which has certain innovation and practical application value. The followings are suggested for authors to improve their manuscript.
- The details of this article need to be re-examined, focusing on issues such as bold words after keywords, different ways of title numbering and miswriting the algorithm name of the organization part in the introduction as April.
- In the literature review, the relevant data of BIM visualization and the corresponding research studies have not been discussed, which are expected to be supplemented.
- In Section 3.1, the picture of the security risk management process framework is mixed with Chinese and English, and the relationship is slightly confused.
- In section 4.2.1, the Bread-related content is not mentioned before and after the article, so what is the significance of Bread? Is it reflected in the table?
- In the 5th section ‘case study’, most of the content is devoted to the establishment of the model. It is suggested to focus on how the model is able to guide twin services such as safety risk assessment, prediction and control in the actual project, while adding BIM visualization results.
- How long does it take to build the model for each project? How long does it take to model each project? Are there many parameters that need to be modified to build the model for different engineering projects? Is it practical?
- Some of the grammar in the article looks strange. Is it obtained by direct machine translation? It is recommended to proofread again. For example, is it more logical to change the first sentence of the abstract to ‘Digital Twin is a method of merging various modern information technologies, and it can be utilized to provide better services for construction projects’. Try to avoid literally translating Chinese into English. In addition, there are imperative sentences in many places in the article, and it is recommended to modify them. Imperative sentences are mostly used in oral expressions and not suitable for formal style. If necessary, they can be used appropriately in the process of discussion, and other places should be avoided as far as possible. It is recommended not to overuse the passive voice, which sometimes makes semantic expression seem unclear.
Author Response
Dear Reviewer,
Thank you for your comments concerning our manuscript entitled “Digital Twin-based Safety Risk Coupling of Prefabricated Building Hoisting”. These comments are all valuable and very helpful for revising and improving our paper, as well as the important guiding significance to our researches. We have studied comments carefully and have made corrections which we hope meet with approval.
We tried our best to improve the manuscript and made some changes in the manuscript. These changes will not influence the content and framework of the paper. And We have marked the modifications in red in the paper.
We have uploaded the reply to your review comments in the form of attachment. In the attachment, we give point-to-point replies to your comments, for example:
Comment 1: The details of this article need to be re-examined, focusing on issues such as bold words after keywords, different ways of title numbering and miswriting the algorithm name of the organization part in the introduction as April.
Response: We are very sorry for our negligence of the details of this article. As for the referee’s concern, we have revised the problem of bold keywords, modified the article number. And the inaccurate algorithm name “April” in the original clause “The fourth section describes the method of coupling analysis using complex network based on mining association rules of April algorithm.” in the organization part in the introduction “2.2. Security risk coupling analysis” has been corrected. The revised sentence is “The fourth section describes the method of coupling analysis using complex network based on mining association rules of Apriori algorithm.” The revised sentence is on line 74-75, page 2 of the revised manuscript.
We appreciate for your warm work earnestly, and hope that the correction will meet with approval. Once again, thank you very much for your comments and suggestions.

Reviewer 2 Report
The authors have presented an interesting paper that is at very good level. However, some key issues have to be addressed by the authors in order to improve the overall scientific merit of the paper. Please find hereafter the recommendations of the reviewer:
- It is suggested to enrich the literature especially on the field of Digital Twins in Section 2.1.
- It is suggested to elaborate the framework and present more technical details as well as the information flow.
- Please describe the communication protocols that have been used for the data transfer between the real and the virtual object.
- All the abbreviations have to be explained within the paper.
- Please check the references format and follow the proposed format.
- It is suggested to minimize the references that are in Chinese language at the minimum necessary level.
- The paper has to be proofread to minimize syntax or grammar errors.
Author Response
Dear Reviewer,
Thank you for your comments concerning our manuscript entitled “Digital Twin-based Safety Risk Coupling of Prefabricated Building Hoisting”. These comments are all valuable and very helpful for revising and improving our paper, as well as the important guiding significance to our researches. We have studied comments carefully and have made corrections which we hope meet with approval.
We tried our best to improve the manuscript and made some changes in the manuscript. These changes will not influence the content and framework of the paper. And We have marked the modifications in red in the paper.
We have uploaded the reply to your review comments in the form of attachment. In the attachment, we give point-to-point replies to your comments, for example:
Comment 3: All the abbreviations have to be explained within the paper.
Response: As for the referee’s concern, the full descriptions of the abbreviations like DT (line 47, page 2), BIM (Line 43, page 2), IoT (Line 44-45, page 2), F-AHP (Line 42-53, page 2), SVM (Line 42-53, page 2), BN (Line 42-53, page 2), ARM (Line 42-53, page 2), IFC (Line 107, page 3) etc. have been supplemented in the revised manuscript.
We appreciate for your warm work earnestly, and hope that the correction will meet with approval. Once again, thank you very much for your comments and suggestions.

Reviewer 3 Report
The authors are focused on presenting of a safety risk management framework for prefabricated hoist-ing based on digital twins. Overall, the paper has a proper structure and the content is interesting. However the paper is difficult to be read mainly due to the language. After language (translation) improvement would be more easy to be formulated observations on the scientific content.
Along the main mentioned problem, the authors can analyze and address some specific issues listed below:
It would be interesting to know how important is to have "self-organizing network". Also IoT is mentioned but the real need for the described algorithm is not clear as stored data is used instead of real-time data. If these parts are essential and were used, a description of the setup can be considered to be presented in a dedicated subsection.
In Abstract:
Unfinished statement "Mining association rules through Apriori algorithm."
Ambiguous phrase: "Refine the complex network characteristics on the Association rule mining(ARM) results, establish the complex network system of the coupling system, and complete the construction of the coupling model.";
Other more suitable term is recommended in: "historical experience data set is -->excavated";
In "1.. Introduction":
- an extra dot is inserted
- The statement is recommended to b revised "At present, the integration of prefabrication and information technology has enabled the construction industry to embark on the road of -->construction industrialization<-."
"In this digital revolution" - there is not clear, which one?
The phrase should be checked in revised: "Each security risk system is frag-mented, and the coupling relationship between different security risk systems, ->systems and factors, and factors and factors<-- cannot be well considered."
In section 2.1:
There is not clear the referred moment of time: "With the advancement of information technology in the -->era<--,"
Unclear statement: "... digital twin technology and -->verified them The feasibility<-- of digital twin technology in risk pre-diction."
In section 2.2
Page 4:
The statement should be revised "... risk coupling theory, and association rule algorithm were first proposed by other fields, and then extended to risk analysis, and have a certain research foundation, but the application in the engineering field is still."
Starting a sentence with a conjunction in English is not common: "And use the complex network to construct a coupling model of the prefabricated hoisting risk complex system ..."
Section 3
Statement should be rewritten as the meaning it is not clear the expressed idea: "Under the active exploration of many scholars, a multi-dimensional and multi-scale application framework of digital twins in various fields has been summarized [18]."
In Fig. 1:
- used text labels should avoid slitting the words on multiple lines (eg. see "Simulatio/n data")
- some parts of the figure ae at low resolution/quality;
In "3.2. ->.<- Information described by coupling model"
an extra sign is present;
Check the usage of capitalization: "The model is as shown in formula (1) -->Shown in [20]:"
There is not clear what authors want to express "VH Bread refers to the virtual information model";
On page 7: there is not clear which are the groups of networks from the statement: "as shown in Figure 3, the form of two groups of networks of ad hoc network equipment is adopted";
Beginning of page 8: "part -->It<- is the storage of coupled";
An extra sign present after "." in the names of the titles for section 4 and subsection 4.1
In Fig. 5 one of the axes (dimension) is not labeled.
Check the capitalization of the 4.2.1 subsection ("4.2.1. data processing");
On page 10: check de punctuation signs usage "The safety risk factor layer is recorded as R -->,< The risk index is defined as n r"
Explanations that follow equations is fine to strat with "where:" instead of "Among them:" for introducing the meaning of the terms.
In Table 5 the limits of intervals should overlap? See limits for s1 and s2 {[0,-->6.243) [-->0.243,13.275)}
On page 12: "The traditional Apriori algorithm has many shortcomings such as ... unique support." can you please explain the idea wanted to be expressed here?
Please improve/review: "Usually introduce a new parameter Lift to improve this", the subject from the statement is not clear;
In subsection 4.2.3
Page 13: Subject of the second sentence is not clear: "Starting from graph theory, build the assembled risk-coupling complex network system." and following "As shown in Figure 8." the predicate is missing.
In "5.2.2. .Construction of coupling model"
an extra sign is present;
Subject missing from statements "Read the divided data set, and mine the association rules among the three risk factors. Select the parameters of the Apriori algorithm ..."
Same issue on page 15: "Calculate the statistical characteristics of the complex network of the coupled system." and 16 "Use intelligent algorithms to mine the association rules among hoisting safety risk factors."
Supplemental sign is present in title of section 6: "6. .Conclusions and Future Works";
Reference:
The first entry can be found inserted between the 3d and 4th item in the list.
The 13th entry is also numbered with 10.
There is too many references in Chinese. One or two wouldn't be a problem but in this case if an international reader want to follow cited references should give up.
Author Response
Dear Reviewer,
Thank you for your comments concerning our manuscript entitled “Digital Twin-based Safety Risk Coupling of Prefabricated Building Hoisting”. These comments are all valuable and very helpful for revising and improving our paper, as well as the important guiding significance to our researches. We have studied comments carefully and have made corrections which we hope meet with approval.
We tried our best to improve the manuscript and made some changes in the manuscript. These changes will not influence the content and framework of the paper. And We have marked the modifications in red in the paper.
We have uploaded the reply to your review comments in the form of attachment. In the attachment, we give point-to-point replies to your comments, for example:
Comment 2: In Abstract.
Unfinished statement "Mining association rules through Apriori algorithm."
Ambiguous phrase: "Refine the complex network characteristics on the Association rule mining (ARM) results, establish the complex network system of the coupling system, and complete the construction of the coupling model.";
Other more suitable term is recommended in: "historical experience data set is -->excavated";
Response: We are very sorry for the grammatical mistake in Abstract. We have recompiled this part according to your opinion. The original sentence “Mining association rules through Apriori algorithm.” And “Refine the complex network characteristics on the Association rule mining (ARM) results, establish the complex network system of the coupling system, and complete the construction of the coupling model.” are amended as follows:(Line 11-14, page 1)
“The real-time perception and virtual-real interaction of multi-source information in the hoisting process are realized, the association rules and coupling relationship among hoisting safety risk factors are mined, and the time-varying data information is visualized.”
We appreciate for your warm work earnestly, and hope that the correction will meet with approval. Once again, thank you very much for your comments and suggestions.

Round 2
Reviewer 3 Report
The work and updates performed by the authors are appreciated but the paper still requires some improvements. Few of points than can be improved are listed below:
line 23-24: "The construction mode relying on a large number of contract employees -->site construction<-- leads to frequent site construction accidents";
line 26: please check the term if it is the intended one: "safety accidents";
line 60: for a reader new in the field of F-AHP, the following statement is unpredictable, for the opposite situation, the reader should guess the correct meaning: "F-AHP is more complicated to calculate the complex evaluation of multi-factor and multi-level."
in line 65 in the current form, the phrase should not end and should continue in 66;
line 69-70: it would be more readable if there would be mentioned the type(s)/nature/meaning of referred data "BIM model is used to visually display the -->data information."line 86: the statement should be checked and linked by adequate punctuation to the preview statement.
line 86: the statement should be checked and linked by adequate punctuation to the preview statement: "And gradually expand the application field.";
lines 95-96: here is really about an "expression" or it is about a model/scheme: "...to realize the -->three-dimensional expression<-- of building safety evacuation...";
line 102: the authors wanted to say about ".... scholars combine and develop BIMs..."?;
lines 105-106: unfinished statement "To improve the efficiency of virtual and real information interaction, and better meet the management needs."
lines 108-109: statement can be slightly updated such that to avoid multiple "and" conjunction usage in one phrase;
lines 114-115: please check and revise such that to avoid the unfinished state of the statement: "For example, combining Radio Frequency Identification (RFID), wireless communication technology [29-31] and intelligent devices [32]."
lines 186-187: please rephrase the statement in order to avoid ambiguity: "In this way, the virtual model and physical site can be interactive and feedback in real-time to optimize and promote each other";
line 195: please make sure that the reader could identify the meaning of the MDT;
lines 199-201: RS term is explained and it is followed by two terms RS_c and RS_s that could induce confusions (they are not present in eq. 1 and not coupled with RS term);
lines 229-232: would be better to update either replace "." after each enumerated item with a dash line ("-"), or enumerate the acquisition terminals and then take each one and explain their role, or that section introduce as a bulleted/numbered list;
lines 319, 331: there is not clear why "Two-states" and "Three-states" are written as proper noun;
in Table 1: rows 3 and 4 column dedicated to the Specific risk indicators join the semicolon from its predecesor element (terms/variable) and follow by a space (eg. "...ratio r15; Wind...");
line 373-374, 376: consider to rephrase from the imperative mode to a case apropriate to the text context: "Take association rule mining for extremely high-risk state ( s1 ) as an example"; and "Try to select parameters in the gentle stage";
In the beginning of the section 5.1 please introduce better the considered scenario (in line 508).
Line 418: please check the statement "The results of data mining analysis -->are do not<-- apply to other engineering projects.";
In case of Fig. 9 how fig. a and b. can be differentiated if the below description are missing?
lines 472-473: please consider to insert the explicit subject in the statement "This completes the construction of the assembled security risk coupling model and maps the coupling results back to the digital twin model.";
Explanations for each equation can start with lower case (eg. ""where" ...) and each explanation can be followed by semicolon instead of dot;
Other recommendations:
- Check again the text of the entire paper as other issues could be found;
- Also to increase the credibility of the research please consider to describe more exactly the data used as their nature and quantities considered. This can be done where you consider apropriate but most suitable can be considered the section 5;
- Include in the conclusions section some remarks to show how obtained results support formulated conclusions.
Author Response
Dear reviewer,
Thank you very much for your kindly comments on our manuscript entitled “Digital Twin-based Safety Risk Coupling of Prefabricated Building Hoisting”. There is no doubt that these comments are valuable and very helpful for revising and improving our manuscript. We have studied comments carefully and have made corrections which we hope meet with approval. We tried our best to improve the manuscript and made some changes in the manuscript. These changes will not influence the content and framework of the paper. And We have marked the modifications in blue in the paper. We would like to answer the questions you mentioned and give detailed account of the changes made to the original manuscript. Our point-to-point responses to your comments are uploaded as attachments.
